# The Added Value of Art for the Well-Being of Older People with Chronic Psychiatric Illnesses and Dementia Living in Long-Term Care Facilities, and on the Collaboration between Their Caregivers and Artists

**DOI:** 10.3390/healthcare9111489

**Published:** 2021-11-01

**Authors:** Petra Boersma, Tjeerd van der Ploeg, Robbert J. J. Gobbens

**Affiliations:** 1Faculty of Health, Sports and Social Work, Inholland University of Applied Sciences, 1081 HV Amsterdam, The Netherlands; robbert.gobbens@inholland.nl; 2Faculty Engineering, Design and Computer Science, Inholland University of Applied Sciences, 1817 MN Alkmaar, The Netherlands; tjeerd.vanderploeg@inholland.nl; 3Zonnehuisgroep Amstelland, 1186 AA Amstelveen, The Netherlands; 4Department Medicine and Population Health, Faculty of Medicine and Health Sciences, University of Antwerp, 2610 Antwerp, Belgium

**Keywords:** artists, dance, visual art, music, interprofessional collaboration, geriatric, positive health

## Abstract

This study sought to provide insight into how art activities influence the well-being of long-term care residents, and how artists and caregivers collaborate in offering these activities. In two long-term care facilities for people with dementia and one for older people with chronic psychiatric disorders, an uncontrolled pre- and post-test study was conducted using a mixed-method design. Forty-six residents participated in the study. Three art activities—(a) dance, (b) music and movement, and (c) visual arts—were studied and co-created with the residents and executed by artists and caregivers together in eight to ten weeks. The Face expression scale (FACE) was used to examine the extent to which participating in the art activity influenced resident mood. Qualitative data were collected via group discussions with artists, caregivers, residents, and an informal caregiver. The results indicated that participating in an art activity positively influenced resident mood (*p* < 0.000). *p*-values for the three art activities were: *p* < 0.000 for dance, *p* = 0.048 for music and movement, and *p* = 0.023 for visual arts. The qualitative data revealed that joining an art activity provided a positive effect, increased social relationships, and improved self-esteem for residents. The collaboration between artists and caregivers stimulated creativity, beauty, and learning from each other, as well as evoking emotions.

## 1. Introduction

‘Art causes emotion; art gives pleasure; art brings people to life’. These are some expressions of how arts can touch people [1]. Young people and (frail) older people can be touched by arts. A recent report by the World Health Organization [2] states that the arts have a major role in prevention of illness, promoting health and the management and treatment of illness across the lifespan. In recent years, art programmes have increasingly been offered to frail older people living in a long-term care facility, especially those living with dementia, Parkinson’s disorder, or psychiatric diseases such as schizophrenia or depression. These art programmes have been offered with the primary goal of creating meaningful personal experiences [3]. Art programmes may also satisfy the sense of aesthetics [4], which is a subdomain of quality of life [5]. Although the evidence base for the effects of art programmes on older people with dementia or people with psychiatric diseases is still relatively small, research suggests there are benefits for quality of life [6,7,8].

In the care for people with dementia or chronic psychiatric disorders, there has been an increased use of various forms of art, such as theatre [9], painting, drawing, singing, dance or sculpture [10,11], music [12,13], and visiting museums [14]. Positive effects were found on different quality of life aspects [9,12,15]. One study found improvement in cognition or decreased apathy or depression [11]. The severity of dementia and the cognitive impairments due to dementia influenced appreciative and active responsiveness and social interaction during the museum visit [14] and could explain why positive study effects are not found. Using arts (music and dance) with people with psychiatric diseases positively influences their mood, anxiety, and social cohesion [8,13], and [7] also lowered levels of negative symptoms.

One of the recommendations of a Dutch review was to invest more in research into art activities and make the responsive elements transparent [1]. The new concept of “Positive Health” has been described by the World Health Organization (2019), in which “Health” is redefined as “The ability to adjust and exert agency in light of life’s physical, emotional, and social challenges” [16]. The Positive Health model assumes a broader understanding of health that contains, for example, quality of life or participating in meaningful activities. It looks at positive effects on six dimensions: quality of life, meaningfulness, mental well-being, participation, daily functioning, and bodily functions [16]. Art activities focus on the dimensions of Positive Health that do not fall within the traditional interpretation of the concept of health in care, such as meaningfulness, societal participation, and quality of life. These concepts from the model of Positive Health in particular show the added value of art activities for frail older people [17].

In 2012, the first national programme in the Netherlands was set up to stimulate participation of older people in art activities. Until that time, welfare and care facilities seldom offered art practice for their elderly residents, and the own initiatives of older people were not yet optimally stimulated and facilitated [18]. The same movement also appeared internationally; art programmes were increasingly offered to frail older people, with the primary goal of creating meaningful personal experiences for the participants [3]. In 2014, the Dutch national policy deployed a stronger connection between cultural and other social domains, including healthcare, which resulting in a major movement towards using arts in health and well-being [1]. Based on various (national) incentives, including from the cultural domain, there have been opportunities in the last few years to develop cultural interventions for people living in a long-term care facility. At the same time, the sustainability and quality of cultural interventions are under pressure due to cutbacks and a lack of structural policy and financing. Healthcare organisations with a structural policy on cultural interventions are scarce in the Netherlands [1], and the beneficial impacts of art activities in healthcare could be furthered through promoting art activities and supporting cross-sectoral collaboration [2]. To our knowledge, this cross-sectoral collaboration (also called interprofessional collaboration) between the care and cultural sectors has not yet received much attention in health research. Interprofessional collaboration is defined as ‘a process in which sharing and working with others towards a common goal is at its core’ [19]. It is a complex and dynamic process that requires specific skills, shared goals, and extra effort [19]. The present study investigated how the interprofessional collaboration of offering art activities in long-term care facilities could be shaped between the artists, the activity counsellors (hereafter ‘caregivers’) who work in the long-term care facilities, and their clients and examined the contribution of both disciplines to the execution of the art activity. The study results contribute, on the one hand, to composing individually tailored art programmes that optimally impact the well-being of older people, and on the other hand, contribute to the optimal execution of art programmes by two disciplines from different organisations.

In present study, we explored this theme as part of a project financed by a national stimulus programme, the development of ‘age friendly cultural cities’. The ‘Arts Education Centre [Scholen in de Kunst]’ thus developed five different art programmes in collaboration with four institutions for elder care and the municipality of Amersfoort. Each programme focused on a specific artistic discipline—(a) dance, (b) music and movement, (c) visual arts, (d) stories, and (e) theatre—and had an artistic value that could contribute to the health and well-being of older people with dementia or chronic psychiatric disorders who live in one of the participating long-term care facilities. The art activities were developed in co-creation with the participants and executed by the artists and caregivers together. The artists in this project were people who had completed an arts education and have also completed teacher training, which allowed them to transfer their art form to others, including children and adults. This project was not so much focused on ‘transferring’ the art form, but rather on providing the elderly with dementia and/or a chronic psychiatric disorder with an opportunity to participate in a form of artistic expression. To tailor the programmes to the personal needs and interests of the participating older people, they were involved from the beginning in the development of the art programmes. The aim of this project was to stimulate the participation of frail older people in society by offering structural cultural interventions, and at the same time, to stimulate the structural embedding of programmes with an artistic value and a preventive or curative effect for older people living with dementia or chronic psychiatric disorders in a long-term care facility. In this study, the programmes (a) dance for people with psychiatric disorders and (b) music and movement or (c) visual arts for people with dementia were investigated.

The objective of this study was to gain insight, on the one hand, into how the art activities—(a) dance, (b) music and movement, and (c) visual arts—influenced the mood and quality of life of older people living with dementia or chronic psychiatric disorders in a long-term care facility, and on the other hand, how artists and professional caregivers collaborated in offering these cultural interventions. We hypothesised that the art activities would positively influence the quality of life and mood of the participants. 

## 2. Materials and Methods

### 2.1. Design

With a pre- and post-test study without a control group, we investigated the extent to which participating in the art activity influenced the mood of older people with chronically geriatric disorders. Qualitative research was also carried out with one individual and three group interviews to discover to the extent to which the art activities contributed to the quality of life and mood of elderly people with a chronic psychiatric/psychogeriatric disorder and also to determine how the collaboration between the artists, caregivers, and clients was designed and what this collaboration yielded.

### 2.2. Setting and Participants

The research was carried out in three wards of three different long-term care facilities in the Amersfoort region in the Netherlands; one ward housed 51 people with chronic psychiatric disorders live and two wards housing 30 people with dementia (14 in one ward and 16 in the other).

### 2.3. Recruitment

All residents of the three wards were invited to participate in the art activity. Subsequently, all residents and in case of the people with dementia, their legal representatives, were asked by the caregivers and artists to participate in the study. The only exclusion criterion was that attendees did not give permission to participate in the study.

### 2.4. Interventions

All three interventions were tailored to the participants’ needs, and there were opportunities for personal input from the participants. By keeping the programmes flexible and working on the basis of improvisation, the artists and caregivers were able to respond to the mood and needs of the participants. There was room for fun in all art activities, but also for feeling sadness. The three interventions were executed by professional artists in collaboration with and with the support of the professional caregivers. The caregivers knew all of the clients and were specifically tasked with handling any difficult feelings and emotions. The artists were not educated as therapists, and the caregivers were the activity counsellors and nurses. Although the artists were not ‘therapists’, the goal of the interventions was to create a setting in which dance, music, and visual arts involved listening to and playing with those in trouble, with the aim of helping them to understand and resolve their predicament. In this, we followed the definition from Brown and Pedder [20] about ‘what is therapy’. Table 1 describes the characteristics of the three art activities.

The dance sessions with people with psychiatric disorders and their caregivers were executed in the central hall of the care facility. During the sessions, the artist and caregivers taught intuitively and sensitively, feeling what the participants needed at that moment. Tempo and energy were alternated throughout the sessions, and up-tempo music was always followed by something light. The last dance session was an open dance class in which family, friends, and other residents could participate.

The purpose of the ‘music and movement’ with people with dementia was to provide a meaningful activity in which interaction, sense of aesthetics, memories, sense of matter, humour, sadness, cognition and conviviality were given a place. The ‘visual arts activity’ with people with dementia and their informal caregivers worked on decorating and filling in a physical cupboard, called ‘Our Cupboard’. In addition to handmade decorations and works of art, a digital photo frame was also added, as well as a black Bakelite phone which told the story of a resident.

### 2.5. Procedure

All art activities were offered over a maximum of ten weeks. If the participants or their legal representatives gave their permission, their mood was measured by the Face expression scale (FACE) [21] on T0 (before participation in the activity) and T1 (immediately after the activity) during select weeks. Data were collected from the participants in the dance activity during the last five weeks, in the music and movement activity during the last three weeks, and in the visual art activity during the last two weeks of the project. Patient characteristics surveyed at T0 were age, gender, illness, disorder duration and, in the case of dementia, the Geriatric Deterioration Score (GDS) [22].

In addition to the quantitative data, qualitative data were collected via four group discussions and one individual interview. A group discussion was held for each artist/caregiver pair (i.e., three total discussions, one each for the dance activity, the music and movement activity, and the visual arts activity). A group discussion was also held with three elderly people who participated in the dance activity. An individual interview was held with the informal caregiver of a person with dementia who participated in the visual arts activity. No interview was held with an informal caregiver for the music and movement activity, because none were present during the activity. All data were collected between 1 May 2019 and 1 July 2019.

### 2.6. Quantitative Data

The ‘overall mood’ of the participants was measured at two moments, T0 and T1 using FACE [21], which consists of one question on a five-point scale: ‘Which facial expression/smiley corresponds most closely to your mood at the moment?’ Participants could then choose from five smiley faces that included very sad, sad, neutral, slightly happy, and very happy. The artists and caregivers presented this question to all participants at T0 and T1. The average inter-evaluator reliability of this instrument was acceptable, 0.58 (95% CI, 0.32–0.85) [9]. The average inter-item correlation in this study was 0.37 (range 0.19–0.54) which is optimal for a 1-item scale [23].

### 2.7. Qualitative Data

A total of ten people participated in the four group discussions: three artists, four caregivers, and three participants. An individual interview was held with the informal caregiver of one participant. Prior to these conversations, a guideline (available from the first author) was drawn up so that the same topics were covered in all conversations: how was the activity carried out? How many people took part in the activity? What were the experiences of the artists, caregivers, and participants? How was the collaboration experienced and to what extent did the art activities contribute to the quality of life and mood of elderly people with a chronic psychiatric/psychogeriatric disorder? For this last part, the items of the Qualidem (a validated quality of life observation list [24,25]) were taken as a starting point. All group and individual interviews were recorded and transcribed, and member checks were carried out by artists and caregivers.

### 2.8. Data Analysis

A power analysis was not necessary in this study because we asked all participants in the arts activities to take part. The quantitative data were analysed with SPSS for Windows, version 24 (IBM, Armonk, NY, USA). Depending on the type of data, patient characteristics were analysed with percentages, averages, and standard deviations. To find out to what extent the mood of the participants after participation in the art activity differed from their mood before participation in the activity, the average score of the participants at T1 was compared to that at T0, using the non-parametric Wilcoxon Signed Ranks Test. The Wilcoxon test then determined the order of the differences between T1 and T0, with three possibilities: at T1, there was a positive progress, a negative progress, or the order was the same (no difference in mood between T0 and T1). The test then calculated the significance level. This two-sided test was performed with an alpha level of 0.05.

The verbatim transcripts were read and analysed deductively using the items of the Qualidem [24,25]. To ensure the coding was carried out as reliably as possible, two of the five transcripts were first analysed independently by two researchers (PB and RG). The remaining three transcripts were then coded by one researcher (PB). All codes were entered in the software programme MAXQDA, version 11. All text fragments of the transcripts were organised by theme [26]. The themes were based on the subscales of the Qualidem: care relationship, positive affect, negative affect, restless behaviour, positive self-image, social isolation, feeling at home, and having something to do. The results were summarised in tables and discussed by PB and RG. Two project-group members from the Arts Education Centre (MB, TK) also participated in this discussion. Using literal quotes and dialogues, the results are presented anonymously.

## 3. Results

### 3.1. Characteristics of the Participants in Our Study

The dance activity was offered twelve times, in which 30 older people with a chronic psychiatric illness participated. The music and movement activity was offered six times, in which nine residents with dementia took part. The visual art activity was also offered six times, in which seven residents with dementia, informal caregivers, and volunteers took part. Residents were involved in only one of the three art activities. Table 2 describes the characteristics of the 46 participants in our study; age was known for 15 participants, the diagnoses for 34 participants, and education for 21 participants. Stage of dementia was established with the GDS [22], which ranged from 1 (no cognitive decline) to 7 (very severe dementia). 

### 3.2. Influence of the Art Activities on the Mood of the Geriatric Participants

The three art interventions together yielded a significant improvement in the mood of the clients after participation (*p* < 0.000). Further analysis per art activity showed that dancing led to a significant improvement in mood (*p* < 0.000). The activities music and movement and visual arts showed a positive trend for significant improvement in mood after the art activity, *p* = 0.048 and *p* = 0.023, respectively (Table 3).

### 3.3. Qualitative Results

Analysis of the four group interviews and the individual interview clarified:how the art activity affected the clients’ quality of life;what the added value was of the collaboration between the artists and the caregivers of the care facility.

#### 3.3.1. Influence of the Art Activity on the Client’s Quality of Life 

According to the participants, artists, and (informal) caregiver(s), the client’s participation in the art activity led to a more positive mood. Analysis of the conversations showed three themes concerning quality of life: positive feelings, social relations, and positive self-image.

Positive affect (feelings)

The artists and caregivers for all three art activities and the clients who took part in the dance activity indicated that participation brought joy. Clients enjoyed participating in the activity. Caregivers saw that the clients were in a good mood after the activity and seemed more relaxed.

Social relations

For some residents, participation led to pleasant contact with co-residents, but usually the contact took place between the client and the artist or caregiver. Respondent 202 expressed: “He is very happy with the rock music. The Rolling Stones, etc. Solid music. Contact. He loves to play. […] He has humour. He feels like making jokes and stuff.”

During the dance activity there was also regular physical contact between the resident and the artist, which gave the residents the feeling of being seen and being recognised. See the dialogue below between the artists and a caregiver:

Respondent 103: “Of course nothing at all is done with whatever need. It is almost taboo.”

Respondent 101 replies: “I sometimes hear: ‘Oh, how beautiful you are!’ And then I think, and I say, ‘Oh, how lovely, thank you.’ ‘Or I’ll get a red head right away’.”

Respondent 202 responds: “I always say: ‘I like you too. I think you’re a nice guy too’.”

Positive self-image

Participation in the activity stimulated the self-esteem of the client—that is, the feeling that they could do more than they thought, and that they could learn something new. The artists and caregivers saw, and the clients themselves indicated, that they could be free of their worries during the activity. One caregiver (respondent 101) reported: “It is very important that people can have a break from being sick for a while. It is so nice to see that people can let go of their worries for a moment and are busy with a primary need. I think it is very important to do that once in a while. Otherwise it’s always about the illness that someone has.”

In conversation with the clients, they said:

Client: “Yes, you forgot your worries.” (respondent 12)

Client: “Yeah, I’ve got that, too, but not so strong.” (respondent 11)

#### 3.3.2. Added Value: Collaboration with Artists and Caregivers

During all group discussions between the artists and caregivers, it emerged that their collaboration was of great added value. The added value was visible in three themes: stimulating creativity, using expertise, and creating beauty/disturbance.

Stimulating creativity

An attempt was made in all activities to match the personal interests of the clients, who were challenged to use their imagination. At the start of the art activity, first, personal interests were identified. For example, in the visual arts activity, clients were asked about their hobbies, such as knitting, cutting figures, and painting. They then used these to fill in ‘images’ in an artistic way in the Cupboard. Respondent 104, a caregiver, expressed this: “I think that an artist has more ideas or that his or her ideas are a bit larger or wider.”

Using expertise

The collaboration between the artists and the caregivers made everyone’s expertise clear and they indicated they learned from each other. The caregivers understood the art of letting the process take its course during the activity. Respondent 206 (artist) greatly appreciated this support: “In the beginning it was very nice that you (caregiver) were there. I was a bit uncomfortable at that time. […] And I noticed your lightness. Then I thought: ‘Oh yes, that’s how I should approach it, with humour.’ And a joke and saying something that’s not right at all. A little playful. At first, I thought that was pretty complicated.”

Conversely, the craftsmanship of the artist meant that the activity was taken to a higher artistic level. In doing so, clients were challenged to push their boundaries. The collaboration went according to plan in all three art activities. The artists and caregivers had their own roles, complemented each other and were willing to learn from each other.

Creating beauty/disturbance

During all art activities, clients were sometimes emotional: happy, but also sad. Dancing to/listening to beautiful music, merging into a game and experiencing they could still perform activities such as knitting, brings emotions with it. The dialogue below explains how an artist stimulates this:

Respondent 204: “I have searched for songs that are musically interesting.”

Interviewer: “What do you mean by that?”

Respondent 204: “Well we always have ‘Que sera sera’. That’s a beautiful song. It’s also got something musically beautiful, and not just lalalala.”

Interviewer: “And what is that, that it brings something musically?”

Respondent 204: “That the feeling is a little touched.”

At the same time, both the artists and the caregivers were also touched during the performance of the art activity. Respondent 202 (artist) expressed this as follows: 

“I think it touches me so much because you’re engaged with each other on a feeling level. While with other groups [of healthy people] I am more concerned with the execution of the movement, the dance technique. Then I say: This should be a little bit like this, that should be a little bit like this. But in this care institutions I don’t really talk during the activity. I don’t know anything about these people, but I feel like I know these people really well. Much more an emotional bond than a dance-technical bond.”

## 4. Discussion

This mixed method study showed how art activities offered to elderly people living with chronic geriatric illnesses such as schizophrenia and dementia and in a long-term care facility positively influenced their mood and quality of life, although the number of participating people with dementia was very small. The qualitative part of this study also showed how caregivers and participants collaborated together. Artists and caregivers learned from each other in terms of expertise, artists stimulated the artistic level to a higher plane, and caregivers in turn stimulated the process to take its course during the activities. From the participants, the caregivers and artists learned which types of music brought beauty and evoked emotions.

Although we have to interpret our findings with caution, we found that older people with schizophrenia or dementia, both living in a long-term care facility, experienced significantly more positive moods after participating in an art activity. Our qualitative data supported this positive change in mood and also showed a positive impact on the quality of life of people with chronic psychiatric disorders and dementia. They reported more positive feelings, more social relations, and a positive self-image. These findings match those of a review of art interventions [27], which reported that participating in an art activity leads to feelings of increased self-confidence and well-being, building new social relationships, participating in meaningful activities, promoting relaxation, and an increased sense of self-worth. In line with our findings, another review described that singing improved mood and quality of life among people with dementia [8]. A Cochrane review found that, although the results were equivocal, quality of life improved among people with schizophrenia who participated in dance therapy [7]. In our small study, we found no significant quantitative or striking qualitative differences between people with schizophrenia or dementia in their experienced quality of life after participating in an art activity. This is in line with Wang and Agius, who stated that listening to music, playing music, or dancing with music appears to be beneficial to both the individual and to the improvement of social cohesion. They explained this by the nature of music, which is an art form that supports human interactions within society and applies to both people with schizophrenia and people with dementia [8]. The average GDS score of those with dementia was 4.9, nearly 5, which indicates moderate dementia. People with dementia have the same needs as persons without dementia—that is, personal and meaningful contact with other human beings, pleasant daytime activities, company and adequate support [28,29]. They consider these needs as being very relevant to their quality of life [28]. In the first stages of dementia, people are still able to take initiative in contact, communication, and activities, but in the later stages (including GDS 4.9), due to progressive cognitive dysfunctions, they become more dependent on other people [30]. For this reason, it is not inconceivable that the stage of dementia affects their perceived mood and quality of life. However, we were unable to determine this in our mixed method study. Despite the ambiguous results, three mental health researchers stated that participating in an art activity can function as a kind of social ‘medicamentation’ which can be used as a supplement to traditional treatments for poor mental health and promoting social engagement for socially marginalised groups, such as people with dementia and people with chronic psychiatric disorders living in a long-term care facility [27,31]. Their statement aligns with the model of Positive Health that our art activities positively influence, including bodily functions, mental well-being, quality of life, and social participation of people with chronic psychiatric disorders and dementia, which are four important pillars of the positive health model [16].

In our study, the art activities were developed in co-creation with the people with dementia and the people with chronic psychiatric disorders. For example, the visual artist inventoried beforehand what activities and personal interests the people with dementia had. She then asked the participants who liked knitting to knit a small square, a participant who liked painting to paint a pattern on wood, and a man who used to be a bread baker to make forms out of dough. The Dutch review mentioned that art activities which are developed in co-creation with the participants seem to more effectively meet the needs and the (life) wishes of the participants and also to better activate the participants [1]. Participant involvement from the first development phase onwards seems to be a condition for the success of a project. The three art activities that were part of our study were executed according to plan and all participants, as well as the artists and caregivers, enjoyed the activities, as was also shown by diverse direct quotations.

The artists and caregivers came into contact with each other through this study. Despite their different backgrounds, they enjoyed the interprofessional collaboration and learned much from each other’s experiences and disciplines. The characteristics of interprofessional collaboration can be described as follows: “an evolving interpersonal process; shared goals, decision-making and care planning; interdependence; effective and frequent interpersonal communication; evaluation of team processes; involvement of older adults and family members in the team; and diverse and flexible team membership” [32]. In our study, the artists and caregivers reported this type of interpersonal process, having shared goals, effective and frequent interpersonal communication, and involvement with the participants as important attributes. For effective interprofessional collaboration, it is necessary to be aware of each other’s role, to have interprofessional education, experience trust between team members, share a belief that interprofessional collaboration improves care, and have organisational support [32]. Except for interprofessional education, the professionals in our study experienced these antecedents and, in line with a study about implementing a visual art programme, the professionals were eager to learn from each other and learn from the art activity [33]. Interprofessional collaboration improves the knowledge of the professionals, their confidence, and their job satisfaction [32].

Looking back on this study, we must also mention some important limitations. The study population was relatively small (*n* = 46), and most participants participated in the dance activity (*n* = 30). This group showed a significant improved mood after the activity (*p* < 0.000), while the participants of the activities music and movement and visual arts showed a tendency to significance (*p* = 0.048 and *p* = 0.023). However, independent of the differences between the three groups, the results were similar. The qualitative outcomes of the three interventions were also equivalent to each other in terms of number of participants and thus complementary to the quantitative outcomes. Another limitation of this study was that we were not able to include a control group in the study design, which makes the quantitative results of this study more uncertain. The last limitation is that we did not obtain all characteristics from all 46 participants.

The use of mixed methods and multiple measures is recommended for studying an art programme [33]. We used quantitative and qualitative methods and measured the mood of the participants before and after the art activity. Nonpharmacological interventions such as our art activities are still not widely used, as there are still few organisations that invest in the implementation and research of these such interventions [31]. Therefore, all opportunities should be taken to explore these new alternative approaches for people with dementia and people with chronic psychiatric disorders, even if the study design could be stronger. It is also important to keep in mind that art will only work if a certain freedom, playfulness, and unpredictability about the outcomes of the cultural intervention are allowed. This does not fit well with the current evidence-based practice culture in long-term care and support [1].

## 5. Conclusions

We conclude that participation in art activities might have a positive influence on the mood and quality of life of people with dementia or a chronic psychiatric disorder, both important aspects of the Positive Health model. At the same time, this research showed that carrying out an art activity together has, for the paired artist and caregiver, not only great added value for the quality of the execution of the activity, but also for the professionals themselves. Interprofessional collaboration between professionals of different disciplines can lead to the execution of the activity being pulled to a higher level: it not only stimulates creativity, but the professionals also learn from each other’s expertise and can jointly bring about beauty and evoked emotions. We therefore recommend long-term care facilities to continue investing in art interventions, particularly those developed in co-creation with the participants, artists, and caregivers. This requires a change in culture towards a less regulation-oriented way of working that is already taking place in the care sector. Artists can contribute to this with their experience and competences.

## Figures and Tables

**Table 1 healthcare-09-01489-t001:** Characteristics of the three art activities.

Characteristics of the Art Activity	Music and Movement	Visual Arts	Dance
Duration	One hour	Two hours	One to two hours
Frequency per week	Once a week	Once a week	Once a week
Aim of the activity	To stimulate the participants mentally and physically by listening to music, making their own music, and using instruments	To develop ‘Our Cupboard’, a project in which people with dementia and their caregivers could undertake and experience ‘making art together’.	To stimulate using physical, cognitive, affective, social, and motor learning objectives. In dance, it is all about the experience that is added to the movement.
Work forms	Singing, moving to music, listening to music, and playing instruments.	Diverse, including, e.g., colouring pictures on wooden panels, working with bread dough, and knitting.	Creativity was challenged using materials, e.g., scarves, balloons, elastic bands, hats, or ribbons. Rhythm and coordination were stimulated using, e.g., plates and drumsticks or clapping and stomping combinations. Contact was stimulated by inviting participants to dance with each other.
Person centredness	Preference of instrument and use of favourite music were asked for or noticed by the artists and caregiver.	During a brainstorming session with the participants, it was discovered what makes them happy, what they like to do. This session was supported with objects and photos and inspired the artist and caregiver to draw up a plan for the Cupboard.	The artist and caregivers asked the participants, or their tutor, for their favourite music. In this way, participants made their own contribution to the session and felt heard and stimulated throughout.
Specifics	Every activity started and ended with the same song	There was no clear approach beforehand, the programme was new, and the artist and caregivers allowed it to unfold.	The dance sessions were easy to follow. The sessions started with a quiet song during which participants ‘arrived’. They were greeted individually by the artist and caregivers and put at ease.

**Table 2 healthcare-09-01489-t002:** Characteristics of the participants (*n* = 46).

Characteristic	Art Activity	*n*	%	Mean (SD)
Project type	Dance	30	65.2	
	Music and movement	9	19.6	
	Visual arts	7	15.2	
Gender	Man	12	26.1	
	Woman	34	73.9	
Mean age (SD) ^b^		15		87.1 (7.1)
Months in care institution	0–6 months	4	8.7	
	7–24 months	5	10.9	
	Longer than 24 months	37	80.4	
Diagnosis ^b^	Schizophrenia	23	50.0	
	Dementia (Alzheimer’s disease)	10	21.7	
	Other	1	2.2	
Education ^b^	Primary school	11	23.9	
	Middle	7	15.7	
	High	3	6.5	
Mean GDS score ^a^ (SD)		10		4.9 (1.7)

^a^ GDS = Global deterioration scale [22]. ^b^ Not all participants were willing to share their characteristics.

**Table 3 healthcare-09-01489-t003:** Results of the Wilcoxon Signed Ranks Test.

Wilcoxon Signed Ranks Test		*n*	Mean Rank	Sum of Ranks	Z	*p* < 0.05
Three projects total (meanT1−meanT0)	Negative ranks	1 ^a^	20.50	20.50	−5.127 ^b^	0.000
	Positive ranks	36 ^b^	18.96	682.50		
	Ties	9 ^c^				
	Total	46				
Dance (meanT1−meanT0)	Negative ranks	0 ^a^	0.00	0.00	−4.269 ^b^	0.000
	Positive ranks	23 ^b^	12.00	276.00		
	Ties	7 ^c^				
	Total	30				
Music & movement meanT1−meanT0	Negative ranks	1 ^a^	4.50	4.50	−1.977 ^b^	0.048
	Positive ranks	7 ^b^	4.50	31.50		
	Ties	1 ^c^				
	Total	9				
Visual arts meanT1−meanT0	Negative ranks	0 ^a^	0.00	0.00	−2.271 ^b^	0.023
	Positive ranks	6 ^b^	3.50	21.00		
	Ties	1 ^c^				
	Total	7				
	Incomplete ranks	1				

^a^ meanT1 < meanT0; ^b^ meanT1 > meanT0; ^c^ meanT1 = meanT0.

## Data Availability

The data presented in this study are available on request from the corresponding author. The data are not publicly available due to privacy restrictions.

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
