# Peer review of "The Added Value of Art for the Well-Being of Older People with Chronic Psychiatric Illnesses and Dementia Living in Long-Term Care Facilities, and on the Collaboration between Their Caregivers and Artists"

_healthcare, 2021, doi:10.3390/healthcare9111489_

Round 1
Reviewer 1 Report
Thank you for the opportunity to review this manuscript.
The opportunity to investigate the impact of art interventions for Caregivers, Artists and Older People with Chronic Psychiatric Illnesses and Dementia is interesting.
There are some issues that need to be addressed in order for this manuscript to be prepared for publication.
There are weaknesses to the paper, perhaps largely in how it is reported, that would need to be remediated for it to be publishable.
Overall, the title does not fully represent the manuscript.
Why authiors put patients' caregivers first and not patients?
- The title led me to believe it was a study describing the added Value of Art but in the study 60% of partecipants are in Dance group.
- I have a concern with the use of the word "Artists." I think it creates confusion. Who are Artists? Are they musicians, dancer etc? Or are music therapists, dance /movement therapist?
The manuscript lacks a clear focus on who does art-activities. This needs to be clarified.
- Line 152-154: For authors the pourpose of ‘Music and movement’ with people with dementia was to provide a meaningful activity in which interaction, sense of aesthetics, memories, sense of matter, humor, sadness, cognition and conviviality were given a place." in my opinion, a certified music therapist is needed to achieve these goals. In discussion (line 343-345) authors report that "music did not improve the mood of people with schizophrenia" citing (n.8) a study of music therapy, and not Music. There is a clear difference in the literature between musical interventions and music therapy interventions.
Many of the studies cited are of music therapy. The authors should clarify this aspect better.
- Would be helpful to know if there were any safety measures considered in the case that music listening, dance, visual art brought up difficult feelings or emotions for the participants. if so, who did that?
- 2.6/2.7/2.8 are ok.
- Results.
-table 2 reported partecipants =53.
- Diagnosis are 40 ( 25-14-1). Please check.
- Education 22 . Please Check
-Most patients participated in the dance activity (n=32).Why?
-The three groups are too different to validly support the authors' conclusions.
-Would it be possible for authors to collect other data for music and visual art? If this is not possible, the authors should provide additional elements to support the discussion.
-The results of the manuscript are too focused on the dance intervention.
-Stronger statement about the limits of this study.
-If authors are unable to provide other data on music and visual art, they should consider reviewing the objective of this study.
Reviewer 2 Report
The manuscript presents the results of work on the impact of three art-activities on the mood and well-being of older people with dementia and chronic psychiatric disorders who live in long-term care-facilities. The relevance of the study is undoubtedly high, but there are many comments on the manuscript:
Line 20 or 161: it is necessary to decipher what FACE is
Lines 169-171: “One group discussion with the artist and caregivers of dance, one with the artist and caregiver of music and movement, and one with the artist and caregiver of visual arts.“ - there is no verb in this sentence.
Line 178: “The mood of the participants was measured at two moments, and T1 using the FACE” - please check the meaning of the sentence
Lines 187-188: “ In the four group discussions participated in total three artists, four caregivers and three participants. In each group discussion participated two or three people”. - The text confuses the reader. The second sentence refers to the participation of three patients, and this is already written in the first sentence.
Line 210: “This test was performed on both sides with an alpha of 0.05.“ – what does it mean “on both sides“?
The text for table 2 (lines 224-229) and the data in table 2 itself do not coincide very much. According to Table 2, 53 people took part in the study. At the same time, in lines 224-225 it is written that 57 people took part in the dancing lessons. Table 3 shows that 30 people were involved in dancing. In addition, when considering Table 3, we can conclude that the participants for different art-activities were different. If so, how could it happen that 40 people took part in the music and movement activity? From all this, the question arises - how many people were engaged in dancing and other activities and were they also engaged in other art-activities or not?
Line 225: “On average, 31.3 clients participated per dance activity” - How should the reader understand this phrase? Should the reader think that, on average, 31.3 participants attended each session?That is, many participated irregularly.If so, then it is necessary to show the dependence of the results on the frequency of attending classes.
Other comments on Table 2:
- Table 2 indicates that the median age was 18 years.Can you explain please?
- Months in care institution is shown only for 52 participants
- Diagnosis is given for only 40 participants
- Education is described for only 22 participants
- Mean GDS score is shown as 15 – The GDS scale does not have such values.Should the reader understand that only 15 participants were tested on the Global Deterioration Scale?The article does not indicate or even describe in general terms what values the participants had on this scale. Authors should describe the level of dementia of the participants.
Neither the Methods nor the Results indicate whether all patients participated in only one of the three art-activities, or did some of them participate in two or three art-activities? if so, it must be described.
The following comments have arisen to the Таble 3: According to the data in Table 2, there were 53 participants, and according to Table 3, there were 46 participants. This requires clarification.
The presentation of dialogues with selected participants when describing the results of the assessment of Qualitative results looks strange in a scientific article. Authors should summarize the results obtained and present them in a form generally accepted for scientific and not literary publications.
In the discussion, the authors did not discuss the differences in outcome between participants with schizophrenia and those with dementia.
The paper does not discuss how the results depend on the stage of dementia.
Round 2
Reviewer 1 Report
Dear authors,
thanks for considering my comments.
the paper is now more detailed and less confusing.
Reviewer 2 Report
The authors responded to the comments and made a number of corrections to the text. However, there are still several issues that should be further corrected before the publication of the manuscript.
Title: The title of the article looks strange, as if Caregivers and Artists were also studied for the influence of art on them.
Lines 24, 55: art activitiesart activities
Line 126: disorders., Qualitative
Lines 179, 198: “four group discussions” – should it be written “three group discussions” ?
Lines 234-235: “The dance activity was offered twelve times, in which 34 (59.6%) women and 23 234 (40.4%) men participated. On average, 31.3 residents participated per dance activity. “ - This text confuses the reader.Probably this text should be deleted. The authors write that 57 people participated in the dances, and then in Table 2 they give information about only 46 participants, and this is for all types of activity.Why this happened is not clear.Besides, according to this text, 31.3 participants on average danced, but according to Table 2, only 30 participants danced in total.This is not clear.
Lines 242-243: please be consistent and if you write “the diagnoses were known for 34 participants”, it should also be described in the text that information about education was received from only 21 participants. In table 2, the rows Mean age (SD) and Mean GDS score (SD) in the Value (%) column do not show percentages, as in the other rows. This makes it difficult for reader to understand the presented data. It might be better to make a separate (additional) column of data for these lines and sign it as the average value, and sign the column with the percentages as percentages.
